# Health Literacy Environment of Breast and Cervical Cancer among Black African Women Globally: A Systematic Review Protocol of Mixed Methods

**DOI:** 10.3390/ijerph17093158

**Published:** 2020-05-01

**Authors:** Lillian Mwanri, Hailay Gesesew, Vanessa Lee, Kiros Hiruy, Hyacinth Udah, Ru Kwedza, Tinashe Dune

**Affiliations:** 1Public Health, College of Medicine and Public Health, Flinders University, Adelaide 5042, Australia; hailushepi@gmail.com; 2Epidemiology, School of Health Sciences, Mekelle University, Mekelle 231, Ethiopia; 3The University of Sydney, University Centre for Rural Health, 61 Uralba Street, Lismore NSW 2480, Australia; vanessa.lee@sydney.edu.au (V.L.); ru.kwedza@health.nsw.gov.au (R.K.); 4Centre for Social Impact Swinburne, Faculty of Business and Law, Swinburne University of Technology, Melbourne 3122, Australia; khiruy@swin.edu.au; 5College of Arts, Society and Education James Cook University, Queensland 4811, Australia; h.udah@griffith.edu.au; 6School of Health Sciences & Translational Health Research Institute, Western Sydney University, Sydney 2751, Australia; T.Dune@westernsydney.edu.au

**Keywords:** Black African women, health literacy, breast cancer, cervical cancer, refugees, protocol, mixed methods

## Abstract

Adequate health literacy is a necessity to enable effective decision making to seek, access and utilise appropriate health care service. Evidence exists indicating a low level of general health literacy among Black African women, especially those with a refugee background. Breast and cervical are the most common cancers, with Black African women or women with African ethnicity being disproportionately overrepresented. The level of health literacy specific to breast and cervical cancer among Black African women, especially those with a refugee background, has not been reviewed systematically. The present study describes a protocol for a systematic review of the available evidence on the level of health literacy specific to breast and cervical cancer among Black African women globally. We will perform a systematic review of the available quantitative and qualitative studies. The search will include studies that describe the level of health literacy specific to breast and cervical cancer among Black African women. We will conduct a preliminary search on Google scholar to build the concepts for search terms, and a full search strategy using the identified concepts and keywords across four databases namely PubMed, SCOPUS, CINAHL and Web of Sciences. We will use Preferred Reporting Items for Systematic Reviews and Meta-Analyses (PRISMA) to schematically present the search strategy. We will use the standardized Joanna Briggs Institute quality appraisal and selection tool to recruit studies, and the data extraction tool to synthesise the information extracted from the recruited studies. We will be guided by socioecological theory and Indigenous epistemology to synthesise the non-quantifiable information thematically, and pool the quantitative information using meta-analysis, based on the availability of information.

## 1. Introduction

Cancers cause a significant burden of disease in women. In 2013, there were 15 million new cancer cases diagnosed, with breast cancer reported as one of the leading causes of mortality and morbidity among women globally. [1] The incidence of breast cancer over the years has been reported to be disproportionately higher among Black African women [2,3,4]. Additionally, Black African women are known to be at a greater risk for early-onset breast cancer globally, and are often diagnosed with more aggressive and advanced forms [5]. Despite this, evidence demonstrates lower participation rates in ethnic communities [6,7], including Black African women, in screening programs in both their home countries and countries of resettlement, with reported reasons comprising behavioral and cultural factors towards general health checkups [4,8,9]. Further, language barriers and unfamiliarity of the health care system and services preclude them from accessing services, [10,11] including screening for breast and cervical cancers [12,13,14].

Cervical cancer is the fourth most common cancer among women globally, with the majority (80%) of reported cases being women residing in developing countries [15,16]. Cervical cancer is still the leading cause of cancer-related deaths among women in developing countries, especially in Africa, due to lack of access to services, poor availability of the Human Papilloma Virus [17] vaccine and poor awareness around screening service and practices [17,18]. Despite significant effort to provide effective services and improve both clinical and preventative programs for cervical cancer, Black African women in Australia remain at a heightened risk of dying of breast and cervical cancers [5,18].

Health literacy, which enables individuals to make critical health decisions including seeking and accessing appropriate health care/services (including breast and cervical cancer clinical and preventative programs), comprises a range of complex skills and knowledge regarding health and health care [14,19,20,21]. There are several definitions for health literacy in general [22,23,24,25]. For example, Nutbeam et al. defined health literacy as “the ability to define to gain access to, understand and use information in ways that promote and maintain good health” [22]. The scope of health literacy measurements involves the following components: health-related attitude, knowledge, behavioral intentions, personal skills and self-efficacy [23,24,25]. Kristine Sørensen performed a systematic review to identify potential definitions and conceptual models—the content analysis of the systematic review produced 12 dimensions of health literacy including knowledge, motivation and competencies of accessing services, and understanding and appraising of health-related information [24].

People with low levels of health literacy have lower rates of health service access and utilisation, poorer health outcomes and higher health care costs than people with higher health literacy [19,26]. There is a growing body of evidence suggesting that health literacy can be a major factor contributing to observed health disparities in migrant communities [2,8,27]. Health literacy specific to breast and cervical cancer prevention, early detection, and therapeutic modalities among Black African women is under-researched globally. Existing evidence shows that Black African women have minimal awareness about screening practices of both breast and cervical cancer [4,16], exacerbated in the country of resettlement by barriers such as language and poor socioeconomic status [28].

Despite the importance of understanding the general link between health literacy and female cancers, there is only one available systematic review published in 2016 which described the link between health literacy and cervical cancer screening, [29] but not specifically for Black African women. As such, the present study will answer the following research question: “What is the available evidence on the level of health literacy specific to breast and cervical cancer among Black African women globally?” As a guide, we will use socioecological theory [30] and Indigenous epistemology (which believes in alternative ways of exploring knowledge construction regardless of ethnic identity or traditional cultures) [31] to synthesise the data and enhance understanding of the level of health literacy specific to breast and cervical cancer among Black African women. Similar to the socioecological theory, Indigenous epistemology provides perspectives, such as relationality and interactions between individual, interpersonal, community, organisational and societal factors, providing a framework for exploring health literacy and helping to meaningfully explain its manifestations and address barriers [32,33]. As such, by using both these theories, we will be able to interpret the multitude of factors influencing the level of health literacy, health behaviors and agency among Black African women. The outcome will inform the development of strategies that address the barriers and their complex interactions and achieve the desired equitable access of care.

## 2. Materials and Methods

### 2.1. Population

This systematic review will include studies on the level of health literacy and access to breast and cervical cancer services among Black African women (aged 25 years old and above) across the world. The selection of this population is based on the fact breast and cervical cancers are common among women aged 25 years and above. Due to lack of consistency in screening globally, we have considered the cervical cancer screening program in Australia, which conducts primary Human Papilloma Virus [17] testing in women aged between 25 and 69 years and exit testing for women aged 70 and 74 years, as a benchmark [34,35]. It is also common knowledge that adult women can develop breast cancer at any age, with higher predisposition with older age [36].

### 2.2. Study Design

We will undertake a comprehensive search of the literature to identify good-quality quantitative and qualitative studies in the English language related to health literacy and access to breast and cervical cancer services among Black African women published between 1980 and 2020. 

### 2.3. Search Strategy

To carry out the search strategy, we will apply the following steps. Initially, a limited search will be performed through Google scholar to develop concepts and key terms for four pre-defined concepts: concept 1 (health literacy, health information, health and wellness, awareness, knowledge or numeracy), concept 2 (breast cancer, cervical cancer, cancer, cancer screening, pap smear), concept 3 (women, reproductive age or girl), and concept 4 (African women, Women of colour, African migrants or settlers, Black women). 

Then, a full search (Appendix A) will be carried out using all identified keywords of each concept and index terms across four databases: PubMed, SCOPUS, CINAHL and Web of Sciences. We will connect concepts 1, 2, 3 and 4 using ‘AND’ and run the full search strategy in these databases. Next, screen titles and abstracts from each database and select relevant titles or abstracts for a full-text critical appraisal. Finally, we will check the bibliographies of relevant articles or other documents. Additionally, we will search the grey literature of unpublished studies from ProQuest Dissertations and Theses (PQDT), health department data, the World Health Organisation (WHO) and other repositories for health data. We will present a schematic presentation of the full search strategy process using the Preferred Reporting Items for Systematic Reviews and Meta-Analyses (PRISMA) guidelines.

### 2.4. Study Selection and Quality Assessment

Quantitative and qualitative primary studies of high quality will be included for review. We will also include systematic reviews if they are available. Studies that used the tools [37,38] described in the ‘Outcome’ section will be included for relevance, feasibility and consistency of the meta-analytic association. Quantitative and qualitative studies that used other validated tools will be included for the narrative review only. Editorials and commentary will be excluded from the review.

Two reviewers, H.G. and L.M., will assess the selected papers for inclusion for methodological validity independently. We will use standardised Joanna Briggs Institute [39] appraisal instruments [39] to appraise the studies (Appendix A). The JBI appraisal instrument consists of nine questions to appraise descriptive and cohort studies critically, ten questions for experimental studies, 11 questions for systematic review studies and ten questions for qualitative studies (Appendix A). We will then calculate the scores for methodological quality for each article using the number of ‘Y’s (yes) as the numerator, and the sum of ‘U’s (unclear), ‘N’s (no) and ‘Y’s as the denominator. We will exclude ‘NA’ (not applicable) from the calculation. If there are any disagreements between the independent reviewers, it will either be resolved by consensus among the research team or through the inviting of a third person. The risk of bias will be assessed via Agency for Healthcare Research and Quality (AHRQ) criteria [40], a tool which evaluates the following biases: selection, detection, performance, reporting and attrition [41]. The tool has separate judgement criteria to assess the risk of bias for each study design and has four values, namely high, moderate, low or unclear risk of bias [42].

*High risk of bias*—significant bias which invalidates the findings, as demonstrated by error in study design, data analysis or reporting.*Moderate risk of bias*—susceptibility of a bias to invalid findings but not enough evidence, as demonstrated by missing data to assess the limitations of the study.*Low risk of bias*—low bias but valid results, as demonstrated by an acceptable allocation of patients to either comparator groups, low attrition (or lost to follow up) rate and appropriate measurement/s of outcome, analysis of data and reporting.*Unclear risk of bias*—difficult to judge but assumed poor reporting of studies.

### 2.5. Data Extraction

We will extract quantitative and qualitative data using the standard JBI data extraction tool (Appendix A). Relevant information will be extracted from all articles included in the review into a spreadsheet containing the year of publication, country (countries) of study setting, study design, study population, sample size, primary and secondary outcomes, and key findings. If reported data are unclear or missing, primary authors of studies will be contacted. The two reviewers will also check the extracted data independently.

### 2.6. Outcomes

The primary outcome of this review will be the level of health literacy specific to breast and cervical cancer among Black African women. Health information literacy will be measured using the Breast Cancer Literacy Assessment Tool (B-LAT) [37] and the Cervical Cancer Literacy Assessment Tool (C-LAT) [38]. The secondary outcome of this review will be on access to breast and cervical cancer services among Black African women.

### 2.7. Data analysis

In the final review, a narrative synthesis of outcomes and the exposure variables will be established. Author’s information, country and/or setting, study population, study design, sample size, and primary and secondary outcomes will be tabulated. Key findings will be outlined to summarise the main information from the included studies. Qualitative information in the included studies will be analyzed using a thematic framework analysis as follows: (i) we will carefully read and understand the relevant data of the included studies, (ii) we will generate key concepts and codes, (iii) we will develop a working analytic framework to produce core themes, (iv) we will map the level of health literacy specific to breast and cervical cancer among Black African women, and (v) we will interpret the mapping results guided by the socioecological theory and Indigenous epistemology. As described above, we will use these theories as a guide to explore and interpret health literacy and gain an understanding of a multitude of factors that exist on the individual, interpersonal, community, organisational and society levels to meaningfully explain health literacy, and related issues, among Black African women.

We will perform meta-analyses to assess the association between the primary outcome and exposures, if data are available from the quantitative studies. Before conducting a meta-analysis, clinical heterogeneity will be checked by the research team, and statistical heterogeneity will be assessed using standard Chi-square and I^2^ tests (I^2^ < 85%, *p* value < 0.05) [43]. A separate meta-analysis will be carried out for the outcome and each exposure using RevMan-5 Software [44]. Forest plots will be used to present the relationship between each exposure and outcome graphically. To calculate pooled unadjusted odds ratio (OR) [45] estimates and their 95% confidence intervals (CI), we will apply random or fixed-effect meta-analysis based on the level of statistical heterogeneity [43]. A fixed-effect model will be considered if there is no heterogeneity, and a random model if there is a moderate level of heterogeneity (I^2^ < 85%). For the meta-analysis, at least two studies should assess the exposure of interest and outcome. A funnel plot will be used to assess the publication bias.

### 2.8. Ethics and Dissemination

Ethical approval will not be required, as primary data will not be collected. Dissemination of the findings will be conducted through journals publications, conference presentations, and a media release.

## 3. Conclusions

This systematic review of the literature will provide evidence on the link between health literacy and breast and cervical cancer screening as well as access to breast and cervical cancer services among Black African women aged 25 years and above. Descriptive and inferential statistical analyses will be used to summarise the quantitative data, and thematic analysis will be used to synthesise the qualitative component of the findings. This review will collate evidence that will provide a substantial contribution to improving access to health care services in general, and breast and cervical cancer services in particular, among Black African women and other similar populations. The findings will be disseminated through publications, conference presentations and a media release.

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
