# Peer review of "Health Literacy Environment of Breast and Cervical Cancer among Black African Women Globally: A Systematic Review Protocol of Mixed Methods"

_ijerph, 2020, doi:10.3390/ijerph17093158_

Round 1
Reviewer 1 Report
Basically it is a research proposal for a future study. The researchers should consider the following points to improve their proposal:
- There is a need to clearly define scope of the term 'health literacy'. The scope of the current study is too limited - location of cancer-related facilities and services. It will also be useful to include "health information literacy" aspect of this topic which will include patients' cancer-related information needs, identification of relevant information sources (including people, web, and the social media), evaluation of information quality, utilization of the located information, etc.
- The proposed search strategy for the identification of literature needs refinement - the concept African Women is unnecessary. Similarly, some phrases and terms suggested for the concept 4 need reconsideration - see my comments in the attached manuscript.

Author Response
Many Thanks for your comments. Please find a point-by-point reply to your comments.

Reviewer 2 Report
The topic will be of great interests to the potential audience. However, one will expect to see the actual results of the study.
The topic is very interesting, but there is no actual information about the results of the author's data collection or analysis. As it is intended to be published as a journal article, I feel that the audience would expect to see a more concrete description of the study results. The submitted article can be improved with more substance.
All meta-analysis projects are with mix-methods in nature. If the authors' intention was to propose a new way of conducting data collection/analysis procedure, there should be an evidence on the advantage of the proposed protocol. However, they authors described their choices of processes without much justification. They simply chose to adopt JBI appraisal instruments and AHRQ criteria without explanation. I believe that the result of their proposed study "health literacy environment of breast and cervical cancer among African women globally" will be of greater impact.
The Review Report Form indeed includes the items, "are the results clearly presented" and "are the conclusion supported by the results". If this is a 'protocol' paper, it should suggest that the protocol can be applied to other similar topics, not just one specific title indicated in their paper. To propose a 'protocol', there should be some supporting evidences, too.
Author Response

(The authors gave the same response as above.)

Reviewer 3 Report
Thank you for the opportunity to review this article. Health literacy is an increasingly important topic, but poorly understood. It is particularly important for groups facing health inequities, such as African women. This protocol read well, but there are a few things that I was not clear on. I have raised these below.
Introduction
A good introduction, with clear links to methods and highlights an important issue. Health literacy is a very complex term, and this isn’t fully unpacked. Some of the more prominent reports and authors on health literacy are also not referred to – i.e. Nutbeam, Sørensen, Nguyen, WHO position on health literacy. Since health literacy is one of your outcomes, I would like to see a bit more thinking about how health literacy is measured.
Methods
Including inclusion and exclusion criteria would be useful. Is this only primary studies? Will editorials, commentary, systematic reviews etc be included? Will all countries and study designs be included?
How will the grey literature be systematically searched? Will this be through Google Scholar? What Health Departments will be searched?
Health literacy as a concept does not have a unified definition and is often used inconsistently in the literature. Some mention of how you conceptualise health literacy, and the inclusion criteria for it in relation to selected studies, would be useful. The link between health literacy and access to health services could also be made more clear in both your introduction and methods (i.e. how will you measure this relationship?)
The tools used to assess breast cancer and cervical cancer health literacy could be better described. It is also unclear how you will apply these tools to selected studies – is this to every study? Or only to studies that use these tools? How will you approach studies that only report on a few measures, or studies using qualitative approaches? Whilst mentioned in your introduction, how you will apply the socioecological theory and Indigenous Epistemology when synthesising data is unclear. It is also not mentioned in your methods but sounds like it should be foundational to your approach.
The population includes women aged 25 years and above. The rationale for this isn’t clear, and a sentence detailing this would be helpful.
It would be good to describe who is included in your population of African women – i.e. is this only women born in Africa? Or include women with African ethnicity? It is also unclear if this only refers to women residing outside of Africa or includes women in Africa.
Conclusion
You conclude that this systematic review will make a substantial contribution – however dissemination is geared towards academics rather than practice or policy. I would suggest revisiting dissemination plan or rethinking the impact of this systematic review.
‘Health literacy status’ is not a term I have heard before, and I am unclear what this refers to.
Good luck in your revisions.
Author Response

(The authors gave the same response as above.)

Reviewer 4 Report
The protocol appears rigorous but I did not read clear research questions. I would have liked to have known the expertise of the team members- How many? Any librarians? A statistician?
Also, the analysis method needs to provide more detail for how qualitative data will be analyzed. The abstract mentions thematic analysis but the protocol method did not provide detail of how this will be performed.
Minor editing needed.
Author Response

(The authors gave the same response as above.)

Round 2
Reviewer 2 Report
Thanks for providing me the revised version. Considering that the respected journal does accept proposal, Yes, I agree that the manuscript has been significantly improved and now warrants publication in IJERPH.